# Biases and limitations in observational studies of Long COVID prevalence and risk factors: A rapid systematic umbrella review

**Miao Jenny Hua**[1,2]*, **Gisela Butera**[3]*, **Oluwaseun Akinyemi**[1], **Deborah Porterfield**[1]

**1** Office of the Assistant Secretary of Planning and Evaluation, U.S. Department of Health and Human Services, Washington, DC, United States of America, **2** Department of Preventive Medicine, Northwestern University Feinberg School of Medicine, Chicago, IL, United States of America, **3** National Institutes of Health Library, Office of Research Services, U.S. Department of Health and Human Services, Bethesda, MD, United States of America

* miao.hua@northwestern.edu (MJH); Gisela.butera@nih.gov (GB)

## Abstract

### Background

Observational studies form the foundation of Long COVID knowledge, however combining data from Long COVID observational studies has multiple methodological challenges. This umbrella review synthesizes estimates of Long COVID prevalence and risk factors as well as biases and limitations in the primary and review literatures.

### Methods and findings

A systematic literature search was conducted using multiple electronic databases (PubMed, EMBASE, LitCOVID) from Jan 1, 2019 until June 9, 2023. Eligible studies were systematic reviews including adult populations assessed for at least one Long COVID symptom four weeks or more after SARS-CoV-2 infection. Overall and subgroup prevalence and risk factors as well as risk of bias (ROB) assessments were extracted and descriptively analyzed. The protocol was registered with PROSPERO (CRD42023434323). Fourteen reviews of 5–196 primary studies were included: 8 reported on Long COVID prevalence, 5 on risk/protective factors, and 1 on both. Prevalence of at least 1 Long COVID symptom ranged from 21% (IQR: 8.9%-35%) to 74.5% (95% CI: 55.6%-78.0%). Risk factor reviews found significant associations between vaccination status, sex, acute COVID-19 severity, and comorbidities. Both prevalence and risk factor reviews frequently identified selection and ascertainment biases. Using the AMSTAR 2 criteria, the quality of included reviews, particularly the prevalence reviews, were concerning for the adequacy of ROB assessments and justifications for conducting meta-analysis.

### Conclusion

A high level of heterogeneity render the interpretation of pooled prevalence estimates of Long COVID challenging, further hampered by the lack of robust critical appraisals in the

**Data Availability Statement:** All relevant data are within the manuscript and its Supporting Information files.

**Funding:** The author(s) received no specific funding for this work.

**Competing interests:** The authors have declared that no competing interests exist.

included reviews. Risk factor reviews were of higher quality overall and suggested consistent associations between Long COVID risk and patient characteristics.

## Introduction

As the COVID-19 pandemic enters its endemic phase, many questions remain regarding the prevalence and risks factors of Long COVID, which has also been called long-haul COVID, post-COVID-19 conditions and a subset of which are post-acute sequelae of COVID-19 [1, 2]. One of the first systematic reviews published on Long COVID estimated that as many as 80% of COVID-19 survivors have at least one long-term post-COVID-19 condition [3]. While natural and vaccine-mediated immunity have reduced rates of hospitalization and death from acute COVID-19, the number of people who have been infected and reinfected with SARS-CoV-2 continues to grow, and with it, cases of Long COVID [4]. Three years into the pandemic, systematic reviews publish estimates of Long COVID prevalence as low as 6.2% [5] to as high as 50% [6, 7]. Moreover, risk and protective factors such as vaccination and infection from different variants of concern remain underexplored areas of research [2]. Observational studies form the foundation of knowledge on Long COVID-19 prevalence and risks, but comparing and aggregating data poses multiple methodological challenges [8, 9]. Lack of robustness in Long COVID observational studies has been remarked on through a recent systematic review of the pediatric population [10]. The aim of this study is to examine the more abundant research on Long COVID in adults that have already been synthesized in systematic reviews through the lens of an umbrella review. This is a useful method for revealing common biases and limitations in the field by synthesizing the critical appraisals that systematic reviews conduct [11, 12].

The main questions of this review are 1) What are the prevalence and risk factors for Long COVID? 2) What kinds of biases and limitations affect the interpretation of observational studies of Long COVID prevalence and risk factors? Given the ongoing challenges to accurately measuring the burden of Long COVID, our goal is to provide guidance for future research to avoid common pitfalls that can impact the validity of observational and interventional studies.

## Methods

We performed a rapid umbrella review of the evidence following the recommendations of the Cochrane Rapid Reviews Methods Group [13]. A rapid review is an evidence synthesis review which follows the systematic review process, and components of the methodology may be simplified or omitted [14]. This review omitted searches of grey literature and data extraction was performed by a single reviewer, which expedited the review process to under six months without compromising on other areas of a systematic review (e.g., critical appraisal) felt to be crucial to ensuring an unbiased protocol. The review was conducted according to the Preferred Reporting Items for Systematic Reviews and Meta-Analysis (PRISMA) 2020 guidelines (see S6 Table). We followed a review protocol pre-registered with the International Prospective Register of Systematic Reviews (PROSPERO) database, CRD42023434323, with no major deviations.

### Eligibility criteria

Eligible study designs were systematic reviews (SR) with or without meta-analyses (MA), excluding narrative, scoping and non-systematic reviews. In terms of the PICO criteria, the

study population included adults aged 18 years and older; reviews including children were eligible if outcomes were stratified by age. Exposure was defined as acute SARS-CoV-2 infection diagnosed 4 weeks or more prior to Long COVID ascertainment, in conformity with the U.S. federal working definition of Long COVID [15]. Comparators (i.e., controls) were defined according to the individual SR reviewed. We considered Long COVID as any or at-least one patient-reported, clinically presented, or administrative (e.g., ICD-10 codes) outcome associated with Long COVID. Studies that exclusively reviewed the prevalence of conditions with preexisting medical definitions (e.g., diabetes) arising post-COVID-19 were excluded, consistent with the WHO consensus definition of post-COVID-19 condition as a diagnosis of exclusion [16]. The relevant context was Long COVID diagnosed and treated in high income countries, thus only peer-reviewed articles in English were considered.

## Search strategy and study selection

The following three databases were searched from January 1, 2019, through June 9, 2023: Lit-COVID, PubMed, Embase. Full database search strategy can be found in S1 Table. In addition to database searches, secondary searches were performed within Web of Science to identify potential reviews that met the eligibility criteria. We manually screened the reference lists of systematic reviews and searched Google Scholar.

Database search results were imported into a reference manager (EndNote X20; Clarivate Analytics) for deduplication, then uploaded into Covidence (Covidence, Melbourne, Victoria, Australia) screening software to remove additional duplicates. An initial pilot was performed to screen title/abstract and full text articles, and any revisions to the search strategy were recorded. Dual screening of both title/abstract and full text was conducted by two reviewers (MH and OA) independently. Any disagreements were resolved by a third reviewer (DP).

## Data extraction and quality assessment

The data extraction template was piloted on a subset of SRs by two independent reviewers (MH and OA). Data extraction was conducted by a single reviewer (MH) into an excel spreadsheet; the extracted data from two SRs were then randomly selected and reviewed by one adjudicator (DP). The SRs' corresponding authors were contacted no more than two times over the course of two weeks to obtain missing data. Data collected included article identifying information, study type, design characteristics of primary studies, setting, participant characteristics, relevant outcomes and the ROB and limitations. Where possible, characteristics specific to studies/populations informing the subset of relevant outcomes were extracted.

## Risk of bias assessment

The AMSTAR 2 critical appraisal tool for SRs that include non-randomized studies was used to determine risk of bias [17]. AMSTAR 2 evaluates SRs through sixteen domains, seven of which critically impact the validity of the review, including protocol registration before commencement of the review (item 2), adequacy of the literature search (item 4), justification for excluding individual studies (item 7), ROB from individual studies included in the review (item 9), appropriateness of meta-analytical methods (item 11), consideration of ROB when interpreting the results of the review (item 13) and assessment of presence and likely impact of publication bias (item 15). Results from studies that have one or more critical weakness will be considered to have low or critically low overall confidence. Studies were assessed by one reviewer (MH) with blinded validation by a second reviewer on one randomly selected study (DP).

### Data synthesis

A meta-analysis was not undertaken as the included SRs were not sufficiently homogenous in population characteristics and design. For meta-analyses that reported relevant outcomes of Long COVID, we reported prevalence as a percentage with 95% CI and risk factors as and odds ratios (OR) or hazard ratios (HR) with 95% CI. The Higgins $I^2$ estimate of heterogeneity was reported for all outcomes where available. If a meta-analysis was not performed, outcomes were reported as median and interquartile range (IQR) if there were at least 5 studies.

We reported pooled estimates or manually calculated the median and IQR of prevalence for each category of 1) hospitalization status (hospitalized, non-hospitalized, mixed); 2) duration of follow-up (<3 months or ≥ 3 months); 3) use of a COVID-negative control group in the primary studies; 4) vaccination status (completed primary series vs. did not complete primary series); 5) COVID variant (wild-type, alpha/beta/delta, omicron). Risk factor outcomes were reported in accordance with the respective SRs. Due to the large number of risk factors investigated, we only reported pooled outcomes or which included at least 5 studies to calculate median/IQR.

We also conducted a narrative synthesis of the ROB identified by the review of the SR. Summary of the ROB comprised the ROB tool used, the number of included primary studies with high or critical ROB, and most frequent ROBs identified.

## Results

The database search resulted in 3,534 references. The reference list from Web of Science and Google Scholar searches yielded one additional article. The title/abstract screening excluded 2,285 articles and the full-text screen excluded 60 articles, most frequently for not including a relevant outcome (see S2 Table). Fourteen SRs were ultimately deemed eligible (see Fig 1). Eight of 14 SRs reported on the prevalence or cumulative incidence of Long COVID (hereafter, prevalence SRs) and five reported on risk/protective factors (hereafter, risk factor SRs). One reported on both prevalence and risk factors in relation to different COVID-19 variants of concern but did not conduct meta-analyses for either outcome. For the sake of simplicity, it will hereafter be counted among the prevalence SRs.

### Study and participant characteristics

All SRs were published between May 26, 2021 to June 8, 2023, with the most recent primary literature search inclusive of February 10, 2023 [18]. The prevalence SRs included 6 SR/MA, 2 SR, and 1 umbrella review with evidence synthesis of a selection of the primary literature. Many SRs did not provide a complete list of citations of the included studies, so we were unable to disambiguate a unique set of primary studies even after contacting the corresponding authors.

Prevalence SRs included 5 to 196 studies and 1643 to 1,289,044 participants. Among the five risk factor SRs, 6–41 studies of 7170 to 860,783 participants were included. Four conducted meta-analyses for relevant outcomes. The most common study designs of primary studies were cohort studies, followed by cross-sectional, and case-control studies. More information on publication and study design can be found in Table 1 and S3 Table.

Among the prevalence SRs, 3 included adults only [18–20], 5 presented age-stratified outcomes [6, 7, 21–23], and 1 reported median or mean ages ≥47 for all relevant primary studies [24]. Diagnosis of SARS-CoV-2 infection was laboratory-based in most SRs that reported exposure ascertainment, but self-reported COVID-19 diagnosis was considered eligible by at least one SR [6]. Timing of follow-up ranged from 28 to 730 days from time zero, which varied as the point of COVID-19 diagnosis, symptom onset, hospital admission or hospital discharge

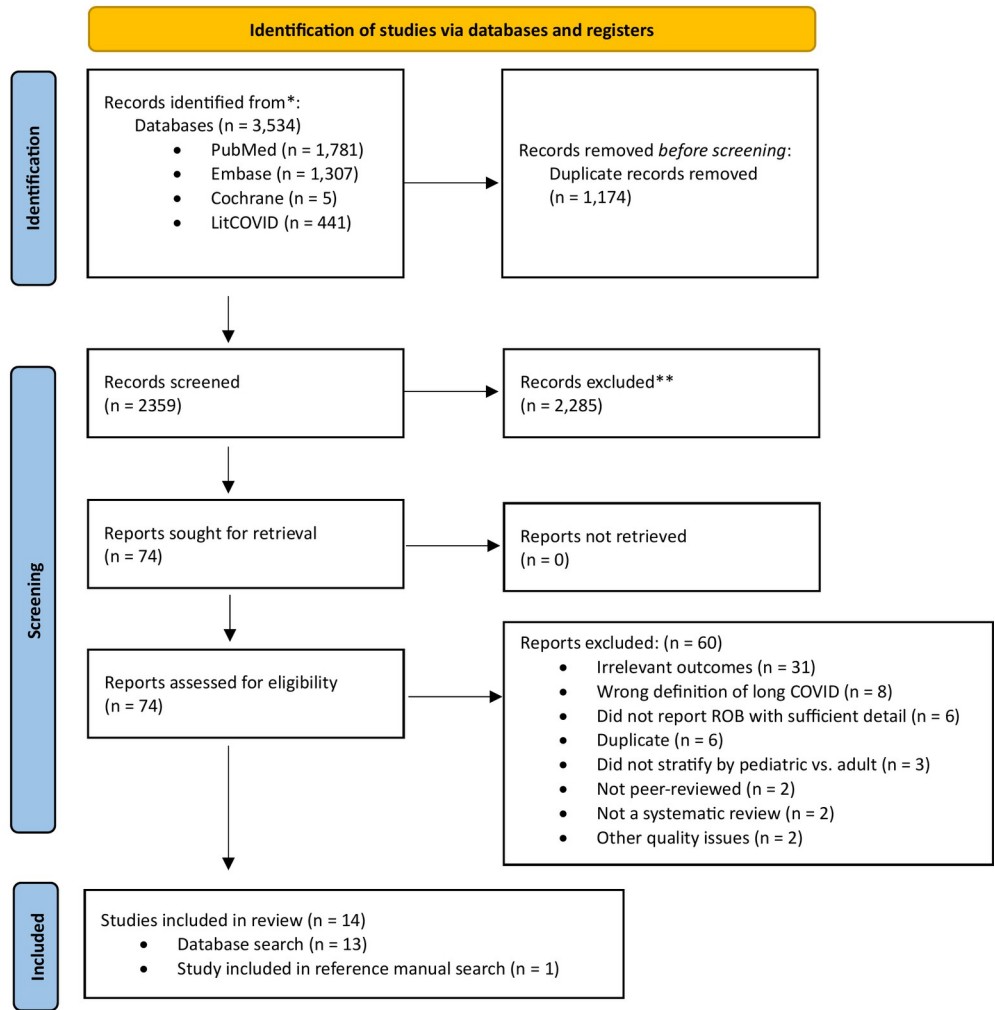

*From:* Page MJ, McKenzie JE, Bossuyt PM, Boutron I, Hoffmann TC, Mulrow CD, et al. The PRISMA 2020 statement: an updated guideline for reporting systematic reviews. BMJ 2021;372:n71. doi: 10.1136/bmj.n71

**Fig 1. PRISMA 2020 flow diagram for systematic reviews.** N, number of studies.

[20, 24]. Of the risk factor SRs, one was restricted to adults [25]; the rest reported mean/median ages of 40–69. Most primary studies in 4 out of 5 risk factor studies confirmed COVID-19 exposure by laboratory methods. For more details on participant characteristics, see Table 1.

## Prevalence outcomes

Long COVID prevalence or cumulative incidence in the SRs ranged from 21% (IQR: 8.9%-35%) to 74.5% (95% CI: 55.6%-78.0%). Hospitalization status was reported by 6 SRs, 3 at the study level (i.e., whether a study included hospitalized patients) and 3 at the individual level for at least one relevant outcome. Of the latter, the percentage of hospitalized patients ranged from 17.4% to 98.2% (see S4 Table). Two SRs conducted random-effects meta-analysis by hospitalization status at the study level [6, 7]. Four provided outcomes that could be stratified by duration of follow-up at 3 months, two of which calculated the pooled prevalence of Long

**Table 1. Study population characteristics, extracted outcomes and assessed limitations and Risks of Bias (ROBs).**

| Study | Type | Population and setting† | Relevant outcome(s) | Number studies & participants in relevant outcome(s) | Limitations Described | Quality assessment of primary studies‡ | Top ROBs |
|---|---|---|---|---|---|---|---|
| **Prevalence SRs** | | | | | | | |
| **Di Gennaro et al. 2022** | SR/MA | Age: Adults and children, mean age 52.3 Sex: 48.8% female Exposure: PCR or lab confirmed Countries/continents: Australia (2), Austria (3), Bangladesh (2), Belgium (2), Brazil (3), Canada (2), China (17), Denmark (3), Ecuador (1), Egypt (3), Faroe Island (1), France (13), Germany (9), Hong Kong (1), India (3), Iraq (1), Iran (3), Israel (3), Italy (35), Japan (3), Korea (2), Latvia (1), Mexico (1), Netherland and Belgium (1), Netherlands (5), Norway (2), Poland (2), Russia (3), Saudi Arabia (1), Singapore (1), Spain (25), Switzerland (6), Turkey (3), UK (13), USA(20) | Outcome 1) Cumulative incidence of any Long COVID signs and symptoms at follow-up from random-effects meta-analysis Outcome 2) Cumulative Incidence of any long COVID signs and symptoms at follow-up from random-effects meta-analysis for mean ages a) 18–60 or b) ≥60 Outcome 3) Follow-up at a) 3 month or b) 3–6 months Outcome 4) By hospitalization a) hospitalized, b) mixed, c) non-hospitalized | Outcome 1) 196 (120,970) Outcome 2) NR Outcome 3) NR Outcome 4) NR | 1) Lack of objective and precise scales for self-report of long COVID symptoms 2) High heterogeneity only partially explained by meta-regression 3) Did not explore role of vaccination 4) Presence of publication bias | Tool: NOS 67/196 moderate ROB (6–7 stars) 129/196 low ROB (8–9 stars) | NR |
| **Fernandez-de-las-Pena 2022 [19]** | SR | Age: adults mean age 50.1 Sex: 56.6% female Exposure: PCR or serological test Countries/continents: Japan, India, Italy, Spain, USA, UK | Outcome 1) Prevalence of at least 1 post-COVID symptom by variants Outcome 2) Risk of long COVID by variants | Outcome 1) 3 (2,115) Outcome 2) 2 (98,104) | 1) High heterogeneity 2) Small number of studies 3) Lack of control for confounders (e.g., vaccination status, reinfections) 4) Many studies investigating long COVID with wild type or Alpha variants sampled hospitalized patients, so may overrepresent chronic fatigue or post-intensive care syndrome due to hospitalization 5) No inclusion of uninfected control groups in any study 6) Cannot exclude potential influence of pandemic-related factors such as social alarm, somatization, physical inactivity, etc. | Tool: NOS 6/6 High quality (7, 7, 7, 8, 8, 9) | 1) Outcome of interest was not present at the start of the study (0/5 cohort/cross-sectional studies) 2) Assessment of outcome was through blind independent assessment or record linkage (2/5 cohort/cross-sectional studies) |
| **Huang et al. 2022** | SR/MA | Age: Adults, 53.3 (mean) Sex: 44.8% female Exposure: NR Countries/continents: Europe, Western Pacific, America, Eastern Mediterranean, Asia and Africa | Outcome 1) Pooled prevalence of any Long COVID symptoms at 1 month by random effects meta-analysis Outcome 2) Pooled prevalence of any Long COVID symptoms at 3 months by random effects meta-analysis Outcome 3) Pooled prevalence of any Long COVID symptoms at 6 months by random effects meta-analysis | Outcome 1) 22 (NR) Outcome 2) 27 (NR) Outcome 3) 20 (NR) | 1) Substantial heterogeneity. Meta-regression found significant association with study region but no other variable (e.g., variable time zero) 2) Prevalence may be over-estimated as original studies included subjects in acute phase and relatively high proportion of severe/critical cases 3) Long COVID symptoms are nonspecific and lack of controls in original studies make it difficult to determine causality 4) Hard to explain temporal change | Tool: NOS for cohort studies (Score interpretation not given) 4/76 3–4 out of 9 stars 53/76 5–7 out of 9 stars 19/76 8 out of 9 stars Tool: AHRQ for cross-sectional studies (Score interpretation not given) 1/5 7 stars 2/5 9 stars 2/5 10 stars | 1) Selection of the non-exposed cohort 2) Comparability of cohorts on the basis of the design or analysis 3) Adequacy of follow up of cohorts |
| **Ma et al. 2023** | SR/MA | Age: Adults, 40–53 (range of means/medians)* Sex: 35–75%* Exposure: Serology or RT-PCR* Countries/continents: Italy, Germany, Brazil and Luxembourg* | Pooled prevalence of at least 1 symptom among asymptomatic SARS-CoV-2 infected adults | 4(99) | 1) Absence of data on asymptomatic cases 2) Absence of some consequences that may not have been recorded | Tool: NOS 4 low ROB (7, 8, 9) | Comparability of cohorts on the basis of design or analysis |

*(Continued)*

**Table 1.** (Continued)

| Study | Type | Population and setting† | Relevant outcome(s) | Number studies & participants in relevant outcome(s) | Limitations Described | Quality assessment of primary studies‡ | Top ROBs |
|---|---|---|---|---|---|---|---|
| **Nasserie et al. 2021** | SR | Age: mean/median range 47–65.5* Sex: 54% male Exposure: PCR or lab confirmation in 14/16, NR in 2/16* Countries/continents: UK (4), Spain (4), China (3), Italy (2), France (1), Canada (1), Austria (1)* | Outcome 1) Median and IQR of prevalence of 1 or more symptom at end of individual or study follow-up Outcome 2) Median and IQR of prevalence of 1 or more symptom at end of individual or study follow-up of a) <3 months or b) ≥ 3 months | Outcome 1) 16 (4,695) Outcome 2) a) 8(1,386) b) 8(3,309) | 1) Design limitations among included studies prevented addressing symptom duration, resolution, and trajectory of global quality of life 2) Symptoms not captured using standardized definitions or instruments too difficult to compare frequency and severity 3) Studies measure same symptoms in different ways report different estimates within the same study 4) Few studies examined past history or baseline prevalence of similar symptoms assessed 5) Variable time-zero: diagnosis or symptom onset, hospital admission, hospital discharge, or recovery from illness 6) Variable follow-up duration 7) Heterogeneity of design features and quality | Tool: NIH (overall summary not performed) a) Patients randomly selected or all eligible patients included 12/16 studies b) Baseline severity reported 11/16 studies c) Attrition 4/16 not reported or ≥30% 3/16 20–29% 6/16 10–19% 3/16 <10% d) Outcome repeatedly measured 2/16 studies e) Established outcome scales used 14/16 studies for some or most outcomes | 1) Repeated outcome measurement 2) Attrition 3) Baseline severity reporting |
| **Nittas et al. 2022** | Umbrella Review and evidence synthesis | Age: Adults and children Sex: NR Exposure: clinical, serological or PCR testing Countries/continents: NR | Outcome 1) Median and IQR of prevalence of at least one Long COVID symptom in adults and children Outcome 2) Median and IQR of prevalence of at least one Long COVID symptom in adult patient studies with population-based samples and adjusted prevalence for cohorts with negative comparators | Outcome 1) 40 (46,144 cases and controls) Outcome 2) 10 (7097 cases, 11,050 controls) | 1) Much early research on SARS-CoV-2 was designed and implemented quickly with a focus on conveniently sampled hospital and outpatient participants so samples recruited early in the pandemic often not as widespread and captured more severe cases; only 4 population-based studies reported prevalence estimates 2) Prevalence of certain symptoms rarely placed in relation to their prevalence in persons without SARS-CoV-2 before or during the pandemic. Most studies fail to distinguish between COVID-related conditions and those linked to preexisting conditions; this is especially true for studies reporting vital organ impairment 3) Certain populations including the elderly, people with disabilities, children and asymptomatic SARS-CoV-2 patients are underrepresented 4) Little evidence on risk and protective factors | Tool: Hoy et al. (overall summary not performed) a) Is the target population representative of the national population? 33/40 high risk 6/40 low risk 1/40 unclear b) Was some sort of random selection used to select the sample, OR was a census undertaken? 36/40 high risk 4/40 low risk c) Was the likelihood on non-response bias minimal? 24/40 high risk 16/40 low risk | Lack of random selection or population-based sampling |
| **O'Mahoney 2023** | SR/MA | Age: 5–73* Sex: 14–94% male (range by studies) Exposure: self-diagnosed or confirmed by a PCR, antigen or antibody test Countries/continents:*: Africa (3), Asia (15), Oceania (1), Europe (53), North America (21), South America(1), multiple (1) | Outcome 1) Pooled prevalence of at least 1 symptom at follow-up by random effects meta-analysis for all studies Outcome 2) Pooled prevalence of at least 1 symptom at follow-up by random effects meta-analysis for studies that included a) hospitalized patients, b) mixed (both hospitalized and non-hospitalized), or c) non-hospitalized Outcome 3) Median prevalence and interquartile range of at least 1 symptom at follow-up time for studies that included only adult participants Outcome 4) Median prevalence and interquartile range of at least 1 symptom at follow-up time for studies that included only adult participants at a) <12 weeks of follow-up or b) at ≥12 weeks of follow-up | Outcome 1) 95 (411,630) Outcome 2) a) 48 (74,422) b) 36 (133,321) c) 11 (203,887) Outcome 3) 83(406,394) Outcome 4) a) 19 (5171) b) 64 (401,223) | 1) 144/194 studies did not report race/ethnicity 2) Limited standardization in using self-report tools 3) Lack of unified consensus definition on long COVID 4) Wide-ranging follow-up period 5) Small number of studies with control/comparator groups 6) Exclusion of studies that recruited from long COVID clinics to avoid selection bias 7) Geographic homogeneity—most studies derive from Europe 8) Did not assess impact of vaccination status and variants | Tool: NIH# Cross-sectional and cohort#: 114 good quality 57 fair quality 14 poor quality Case-control#: 5 good quality 4 fair quality Case series#: 1 good 1 fair | 1) Sample size justification, power description, or variance and effect estimates was not provided (170/184 cohort studies) 2) Exposure was not assessed more than once over time (170/184 184 cohort studies) 3) Key potential confounding variables were not measured or adjusted on the impact between exposure and outcomes (95/184 184 cohort studies) |

*(Continued)*

**Table 1.** (Continued)

| Study | Type | Population and setting† | Relevant outcome(s) | Number studies & participants in relevant outcome(s) | Limitations Described | Quality assessment of primary studies‡ | Top ROBs |
|---|---|---|---|---|---|---|---|
| **Rahmati et al. 2023** | SR/MA | Age: Adults, median age range 40–61* Sex: 28–68% female* Exposure: NR Countries/continents: China (7), USA (1), Spain (1), France (1)* | Pooled event rate of any Long COVID symptom from random-effect meta-analysis | 10(4,589) | 1) Substantial heterogeneity possibly due to small study size, different assessment scales and wide variation in reported prevalence data (e.g., prevalence of at least one unresolved symptom ranged from 16% to 76%) 2) Paucity of data up to 2 years of COVID-19 follow-up 3) Only 1/3 of included studies conducted in-person assessment during follow-up, the rest was mixed of in-person and phone, only-phone or via electronic health record, so risk of recall bias 4) Most studies conducted in China so may not generalize to Europe, USA and resource-poor nations 5) Most study participants were from hospitalized patients during early waves of the pandemic so cannot generalize to new variants or non-hospitalized patients 6) Only a slight majority of studies had control/comparator groups | Tool: NOS 9/10 moderate ROB (7 stars) 1/10 low ROB (8 stars) | 1) Selection of unexposed cohort was from the same community as exposed (1/10 studies) 2) Assessment of outcome was through blind independent assessment or record linkage (0/10 studies) |
| **Zeng et al. 2022** | SR/MA | Age: Adults, 58–68.8 (range of means/medians)* Sex: 54.8% (individual-level)* Exposure: NR* Countries/continents: Spain (3), France (2), UK, USA, China* | Pooled prevalence of at least 1 symptom among SARS-CoV-2 infected adults from cohorts with median/mean age ≥60 | 8(3658) | 1) High heterogeneity between studies with most I-squared >50%, heterogeneity may be due to different case definitions, diagnostic criteria, and follow-up durations 2) Outcome based mainly on self-report and limited objective evidence 3) Cannot ascertain causality 4) Only reviewed work done with Alpha variant 5) Did not involve vaccination because of limited reporting in primary studies | Tool: NOS <5 stars = low quality 5–7 stars = moderate quality >7 stars = high quality Full sample: 12/151 high quality 124/151 medium quality 15/151 (9.9%) low quality Relevant outcome: 1/8 low quality (4) 7/8 moderate quality (5,5,5,6,6,6,7) | NR |
| **Risk Factors SRs** | | | | | | | |
| **Byambasuren et al. 2023** | SR | Age: Individuals eligible to receive any COVID-19 vaccine during study period (only adults in at least 15/16 studies) Sex: NR Exposure: NR Countries/continents: USA (8), UK (4), Netherlands (2), France and Italy | HR or OR of at least 1 long COVID symptoms, most common symptoms, long COVID of any severity, receiving care >3 months after infection, confusion/difficulty concentrating, risk of fatigue after: 1) 1 dose of pre-infection vaccine vs. 0 dose 2) 2 doses of pre-infection vaccine vs. 0 dose 3) 3 doses of pre-infection vaccine vs. 0 dose 4) Any dose of pre-infection vaccine vs. 0 dose Median and IQR given if there are 5 or more studies; OR and 95% CI given if there is only 1 study | 1) 1 dose: 5(NR) 2) 2 doses: 5(NR) 3) 3 doses: 1(318 vaccinated; 421 unvaccinated) 4) Any dose: 5(NR) | 1) Lack of consistent definition of Long COVID 2) Could not recalculate common ratio so used HR/OR/RR depending on study. 3) Could not conduct meta-analysis due to high heterogeneity and lack of data on specific vaccine type, time between exposure and disease and viral variant. 4) Could not determine prevalence of individual symptoms as not reported in most studies. 5) Lack of high quality primary literature, in particular RCTs 6) Less than half of studies used PSM to form comparison group. | Tool: ROBINS-I Assessed adjustment for predetermined confounders: age, sex, BMI, initial disease severity, comorbidity, vaccine hesitancy 3 critical ROB 5 serious ROB 3 moderate ROB | 1) Selection of reported result (serious or critical in 11/16 studies) 2) Confounders not adjusted for (serious or critical in 9/16 studies): vaccine hesitancy, initial disease severity, sex 4) Dealing with missing data (not described or serious ROB in 7/16 studies) 4) Measurement of outcome (serious or critical in 5/16 studies): ICD-10 codes with high detection bias; unclear definition and self-reported outcomes |
| **Notarte et al. 2022** | SR/MA | Age*: 40–65 (range of means/medians) Sex*: 51.9% female (NR in 2/7 studies) Exposure*: RT-PCR Countries/continents*: UK(3), Switzerland (1), Italy (1), Faroe Islands (1), Spain (1) | Pooled OR and 95% CI for the association between sex and presence of any long COVID-19 symptom | 7(386,237 COVID survivors; 1,944,580 COVID-negative controls) | 1) Lack of consistent definition of long COVID 2) Only studies that used WHO definition included in meta-analysis so small number of studies 3) Lack of differentiation in risk factors (i.e., hospitalization status, variants of concern) 4) Did not investigate COVID-19 associated risk factors (e.g., severity of acute infection) | Tool: QUIPS 6 domains: Low ROB if ≤1 domain has moderate ROB; High ROB if ≥1 domain high ROB or ≥3 domains moderate ROB; all papers in between classified as moderate ROB 6/16 high ROB 5/16 moderate ROB 5/16 low ROB | 1) Adjustment for other prognostic factors (2H, 10M) 2) Attrition (4H, 5M) 3) Participation (2H, 2M) 3) Prognostic factor measurement (2H, 2M) |

*(Continued)*

**Table 1.** (Continued)

| Study | Type | Population and setting† | Relevant outcome(s) | Number studies & participants in relevant outcome(s) | Limitations Described | Quality assessment of primary studies‡ | Top ROBs |
|---|---|---|---|---|---|---|---|
| **Pillay et al. 2022** | SR/MA | Age: 42.7–69 (median range)* Sex: 49.0% (median)* Exposure: ≥90% lab confirmed in 12/17 studies* Countries/continents: China (5), Italy (2), Norway (2), Russia (2), Switzerland (2), USA (1), UK (1), Sweden(1) and Turkey (1)* | Pooled OR and 95% CI for non-recovery/persistent systems from random-effects meta-analysis for: 1) Age (continuous) 2) Age (categorical) 3) Sex (female vs. male) 4) Comorbidities (≥1 vs. 0) 5) Acute COVID-19 Severity (Critical/ICU vs. not) 6) Acute COVID-19 Severity (Severe/Critical vs. not) 7) Need for hospitalization | Total: 9(7170) 1) 2(3396) 2) 40–60 vs 18–40: 4 (2867) >60 vs 18-40yrs: 3 (1440) 3) 8(6163) 4) 4(2069) 5) 3(1722) 6) 2(1438) 7) 2(1030) | 1) Findings applicable mainly to long-term consequences ≥22 weeks after acute illness 2) Large proportion of hospitalized population 3) Several potential risk factors but none identified as strong association with long COVID outcomes 4) Evidence sparse on pre-existing socioeconomic variables (e.g., race/ethnicity, income, education, employment) 5) Self-reported outcome/exposure data so subject to recall and misclassification bias | Tool: JBI Checklist for Cohort Studies 5/9 studies high ROB (≥2 domains high ROB) 4/9 some concern for ROB (<2 domains high ROB) | 1) Use of appropriate statistical analysis (2/9) 2) Risk factor measurement in valid/reliable way (4/9) 3) Outcome measured in valid/reliable way (5/9) 3)Follow up complete, and if not, the reasons to loss to follow up were described and explored (5/9) |
| **Tsampasian et al. 2023** | SR/MA | Age: adults (≥18) Sex: NR Exposure: RT-PCR, serology or other laboratory confirmation in 38/41 studies; self-report admitted in 3/41 studies Countries/continents: Europe (30), Americas [Brazil, Canada, US] (6), Asia (5), Africa (1) | Pooled OR and 95% prediction intervals (PI) for developing post-COVID condition from random-effects meta-analysis for: 1) Sex (female vs. male) 2) Age (≥40 vs. 18–40) 3) BMI (≥30 vs. <30) 4) Smoking status (current smoker vs. nonsmokers) 5) Comorbidities a) Anxiety/depression b) Asthma c) CKD d) COPD e) Diabetes f) Immunosuppression g) Ischemic heart disease 6a) Hospitalization (hospitalized vs. not) 6b) ICU admission (admitted to ICU vs not) 7) Vaccination status (2 doses vs. unclear) | 1) 38 (727,630) 2) 9 (324,950) 3) 16 (701,807) 4) 20 (455,204) 5a) 4(634,734) 5b) 13(639,397) 5c) 8(255,791) 5d) 10 (257,340) 5e) 18(259,978) 5f) 3(967) 5g) 5(201,906) 6a) 8(265,466) 6b) 10(213,441) 7) 4(249,788) | 1) High heterogeneity in many outcomes 2) Limitation of NOS scale itself even though all studies were rated as moderate or high quality 3) High ROB associated with observational studies 4) Different definitions of symptoms included among different studies 5) Pooled results independent of variants | Tool: NOS 11/41 moderate quality (6/9 stars) 30/41 high quality (7-9/9 stars) | NR |
| **Watanabe et al. 2023** | SR/MA | Age: 45–58 years (range of medians/means)* Sex: 9.0–63% female* Exposure: PCR, serology or symptoms Countries/continents: UK (2), USA (2), Turkey (1) and Italy (1)* | Pooled OR and 95% CI for the incidence of long COVID after: 1) 2 pre-infection vaccination doses vs. 0 dose 2) 2 pre-infection vaccination dose vs. 1 dose; 3) 1 pre-infection vaccination dose vs. 0 dose) by random effects meta-analysis | 2 doses vs. 0 dose: 4(2 doses: 60,099; 0 dose: 536,291) 2 doses vs. 1 dose: 3(2 doses: 3142;1 dose: 21,872) 1 dose vs. 0 dose: 2 (1 dose: 15,842; 0 dose: 392,745) | 1) Only observational studies included 2) Varied proportion of ICU-admitted patients 3) Definition of long COVID varied 4) Could not evaluate variants of concern as observational period varied widely 5) Could not evaluate effect 3–4 vaccine doses | Tool: NOS 1/5 moderate risk of bias (7) 4/5 low risk of bias (8, 8, 9, 9) | Representativeness of the exposed cohort |

AHRQ, Agency for Healthcare Research and Quality; BMI, body mass index; CI, confidence interval; CKD, chronic kidney disease; COPD, chronic obstructive pulmonary disease; H, high; HR, hazard ratio; ICU, intensive care unit; IQR, interquartile range; JBI, Joanna Briggs Institute; M, moderate; MA, meta-analysis; NIH, National Institutes of Health; NOS, Newcastle–Ottawa Scale; NR, not reported; OR, odds ratio; ROB, risk of bias; ROBINS-I, Risk of Bias in Non-randomized Studies–of Interventions; RT-PCR, reverse transcriptase polymerase chain reaction; QUIPS, Quality in Prognosis Studies; SR, systematic review; WHO, World Health Organization

†: data in column extracted for all studies included in SR unless otherwise specified by asterisk (*); ‡: data in column extracted only for studies included in relevant outcomes unless otherwise specified by number sign (#)

| Study | Outcome | N (participants) | Percent at least 1 symptom, 95% CI | I-squared |
|---|---|---|---|---|
| Zeng et al. 2022 | 1) Age >60 | 8 (3658) | 60.28 [50.47, 69.69] | 97% |
| Di Gennaro et al. 2022 | 1) All studies | 196 (120,970) | 56.9 [52.2, 61.6] | 99% |
| | 2a) Ages 18-60 | NR | 58.0 [51.8, 64.1] | NR |
| | 2b) Ages >60 | NR | 56.2 [47.2, 65] | NR |
| | 3a) 3 months follow-up | NR | 60.7 [49.5, 71.4] | NR |
| | 3b) 3-6 months follow-up | NR | 56.0 [48.8, 63.0] | NR |
| | 4a) Hospitalized | NR | 51.5 [45.0, 58.1] | NR |
| | 4b) Mixed | NR | 55.7 [46.3, 65.1] | NR |
| | 4c) Non-hospitalized | NR | 53.0 [38.5, 67.4] | NR |
| Huang et al. 2022 | 1) 1 month follow-up | 22 (NR) | 55 [47, 63] | 97.80% |
| | 2) 3 months follow-up | 27 (NR) | 52 [39, 64] | 99.70% |
| | 3) 6 months follow-up | 20 (NR) | 54 [34, 73] | 99.90% |
| O'Mahoney et al. 2023 | 1) All studies | 95 (411,630) | 44.8 [38.6, 51.2] | NR |
| | 2a) Hospitalized | 48 (74,422) | 52.63 [43.46, 61.64] | 99.70% |
| | 2b) Mixed | 36 (133,321) | 34.46 [21.86, 49.70] | 99.80% |
| | 2c) Non-hospitalized | 11 (203,887) | 37.80 [31.82, 44.18] | 99.50% |
| Ma et al. 2023 | 1) Asymptomatic COVID-19 | 4 (99) | 21.38 [11.12, 53.87] | 94.30% |
| Rahmati, 2023 | 1) 2-years post-COVID | 10 (4,589) | 41.7 [40.1, 43.2] | NR |

**Fig 2. Meta-analytic estimates of prevalence extracted from Long COVID systematic reviews.** N, number of studies; CI, confidence interval; NR, not reported.

COVID by duration of follow-up [7, 20]. For the rest, we calculated the median prevalence and inter-quartile ranges stratified by 3 months/12 weeks.

The inclusion of COVID-19 negative comparator groups was noted in three SRs; only one estimated the prevalence of Long COVID stratified by the use of control groups [21]. Three SRs reported the SARS-CoV-2 variant of concern assumed responsible for most infections [19, 22, 23], one of which reported Long COVID prevalence by variant of concern without conducting a meta-analysis [19]. Two SRs commented on vaccination status reported in primary studies [7, 19], neither of which reported prevalence estimates by vaccination status. See S5 Table for more information on comparator groups and other subgroups.

Outcomes with pooled estimates are shown in Fig 2. The I$^2$ measure of heterogeneity was over 90% in all reporting pooled prevalence estimates but an I$^2$ was not reported for 50% of pooled estimates. Outcomes with median and IQR are shown in Fig 3.

## Risk factor outcomes

Up to fourteen different risk/protective factors were extracted from the five risk factor SRs reviewed. Three examined the associations between vaccination and Long COVID risk and found that two or more pre-infection doses of COVID-19 vaccine significantly decreased the risk of Long COVID compared to no or 1 vaccine, but a single dose did not significantly mitigate Long COVID risk [25–27]. Three SRs reported on risks associated with age, sex, acute COVID-19 severity and other sociodemographic and clinical risk factors [25, 28, 29]. All three found female sex to be a significant risk factor for Long COVID. One SR found that categorical

| Study | Outcome | N (participants) | Percent at least 1 symptom, Median [IQR] |
|---|---|---|---|
| Nasserie et al. 2021 | 1) All studies | 16(4,695) | 72.5 [55.0-80.0] |
| | 2a) <3 months | 8(1,386) | 69.0 [60.0-77.3] |
| | 2b) ≥3 months | 8(3,309) | 74.5 [55.6-78.0] |
| Nittas et al. 2022 | 1) All studies | 40(46,144) | 49 [24-64] |
| | 2) Adult, controlled or representative | 10(18,147) | 21 [8.9-35] |
| O'Mahoney et al. 2023 | 3) Adults only | 83(406,394) | 51.0 [39.0-63.7] |
| | 4a) Adults, <12 weeks | 19(5,171) | 57.7 [48.3-73.9] |
| | 4b) Adults, ≥12 weeks | 64(401,223) | 50.9 [39.9-62.6] |

**Fig 3. Median and interquartile ranges of prevalence estimates extracted from Long COVID systematic reviews.** N, number of studies; IQR, interquartile range.

age of 40 or greater posed increased risk of Long COVID (OR: 1.21, 95% CI: 1.11–1.33) [25], while another did not find evidence of age-associated risks [28]. The latter also found elevated risks of non-recovery from severe or critical acute COVID-19 with moderate certainty, although associations with hospitalization status were non-significant [28]. Both SRs investigated associations between Long COVID and comorbidities, finding significant risk associations. Full statistical outcomes are summarized in S4 Table.

## Summary of ROB assessments of primary studies

A majority of SRs (6 out of 9 prevalence SRs, 2 out of 5 risk factor SRs used the Newcastle-Ottawa Scale (NOS) [30]. Other tools used in the prevalence SRs included the NIH tool (2 studies) [31], Hoy et al. (1 study) [32] and the AHRQ tool (1 study) for cross-sectional studies [33] in a study that also used the NOS for cohort studies. Risk factor SRs also employed ROBINS-I (1 study) [34], JBI checklist for cohort studies (1 study) [35] and QUIPS (1 study) [36].

Six prevalence SRs gave an overall ROB score. Regardless of the tool used, the majority of primary studies were scored as having low or moderate ROB, with only 0–9.9% of studies rated has having a high ROB (e.g., less than five out of nine on the NOS scale). The risk factor SRs reported higher ROB overall, with 0–56% of primary studies rated as having a high or critical ROB.

Out of fourteen SRs, two prevalence SRs [7, 22] and one risk factor SR [25] reported only the overall ROB of the primary studies without a score breakdown. From the 11 SRs that did report a score breakdown, the quality of outcome ascertainment and selection bias were the most frequently top-ranked ROBs; adjustment for confounders, attrition, representative sampling and outcome ascertainment were areas of deficiency in at least two risk factors SRs.

## Summary of limitations as described in the SRs

High heterogeneity was identified as a limitation in 8 prevalence SRs, with the exception of the SR on asymptomatic cases [23]. Lack of control/comparator group or representative sampling was noted in 5 prevalence SRs. Lack of standardization in case definition and symptom measurement was noted in 3 prevalence SRs and 4 risk factor SRs. Variable follow-up time, lack of data on age, race/ethnicity, disability and overrepresentation of people hospitalized with COVID-19 were also identified as limitations.

## ROB assessment of SRs with AMSTAR 2

All prevalence SRs had weaknesses in at least two critical domains (see Table 2) [17]. Among these, deficiencies on items 9 and 11 are particularly concerning for our aim of identifying biases and limitations in the Long COVID evidence base.

An adequate score on item 11 required the SR to explicitly justify the decision to perform a meta-analysis based on the compatibility of included studies. None of the prevalence SRs that conducted a meta-analysis included such a statement, while all of the SRs which did not pursue a meta-analysis cited the high degree of heterogeneity in the primary literature as deterrent. We considered this item satisfied if meta-analysis was primarily undertaken by pre-determined subgroups that may create comparable cohorts, such as by hospitalization status or follow-up duration. Nevertheless, the subgroup outcomes in prevalence SRs all reported $I^2$ statistics greater than 90%.

ROB assessments conducted by the SRs were often deficient, hence the large number of partial or complete deficiencies on item 9. For a "yes," the SR had to evaluate primary studies on at least four sources of bias: confounding, selection, measurement of exposures and outcomes, and selective reporting of analyses or outcomes. The NOS rates ROB across the three domains of selection, comparability and outcome assessment without any criteria for selective reporting

**Table 2. AMSTAR 2 assessments of all SRs.**

| Reference | 1 | 2 | 3 | 4 | 5 | 6 | 7 | 8 | 9 | 10 | 11 | 12 | 13 | 14 | 15 | 16 | Overall Confidence |
|---|---|---|---|---|---|---|---|---|---|---|---|---|---|---|---|---|---|
| **Prevalence SRs** | | | | | | | | | | | | | | | | | |
| Ma et al. | Yes | Yes | Yes | Partial yes | Yes | Yes | No | Yes | Partial yes | No | Yes | Yes | No | No | Yes | Yes | Critically Low |
| Nittas et al. | Yes | No | Yes | Partial yes | Yes | Yes | No | Partial yes | Partial yes | No | No MA | No MA | Yes | Yes | No MA | Yes | Critically Low |
| Rahmati et al. | Yes | Partial yes | Yes | Partial yes | Yes | Yes | Yes | Partial yes | Partial yes | No | No | No | Yes | Yes | Yes | Yes | Critically Low |
| Di Gennaro et al. | Yes | No | Yes | No | Yes | Yes | No | No | No | No | No | Yes | No | Yes | Yes | Yes | Critically Low |
| O'Mahoney et al. | Yes | Yes | Yes | Partial yes | Yes | Yes | No | Yes | Yes | No | Yes | Yes | Yes | Yes | Yes | Yes | Low |
| Zeng et al. | No | Partial yes | Yes | Partial yes | No | Yes | No | No | Partial yes | No | No | No | No | Yes | Yes | Yes | Critically Low |
| Fernandez-de-las-Peñas et al. [19] | Yes | No | Yes | Partial yes | Yes | Unclear | No | Partial yes | No | No | No MA | No MA | Unclear | Yes | No MA | Yes | Critically Low |
| Huang et al. | Yes | Yes | Yes | Partial yes | Yes | Yes | No | Partial yes | No | No | Yes | No | No | Yes | Yes | Yes | Critically Low |
| Nasserie et al. | No | No | Yes | No | Yes | Yes | Yes | No | Yes | No | No MA | No MA | Yes | Yes | No MA | Yes | Critically Low |
| **Risk Factor SR** | | | | | | | | | | | | | | | | | |
| Pillay et al. | Yes | Yes | Yes | Yes | Yes | Yes | Yes | Yes | Yes | Yes | Yes | Yes | Yes | Yes | Yes | Yes | High |
| Tsampasian et al. | Yes | Yes | Yes | Yes | Yes | Yes | No | Yes | No | No | Yes | Yes | Yes | Yes | Yes | Yes | Critically Low |
| Byambasuren et al. | Yes | Yes | Yes | Yes | Yes | Yes | Yes | Yes | Yes | No | No MA | No MA | Yes | Yes | No MA | Yes | High |
| Notarte et al. | Yes | No | Yes | No | Yes | Yes | No | Partial yes | Partial yes | No | No | No | Yes | No | No MA | Yes | Critically Low |
| Watanabe et al. | Yes | Partial yes | Yes | Partial yes | Yes | Unclear | No | No | No | No | No | No | No | No | Yes | Yes | Critically Low |

AMSTAR = A Measurement Tool to Assess Systematic Reviews; MA = meta-analysis; SR = systematic review

Items 2, 4, 7, 9, 11, 13 are considered "critical." Studies that have one "no" in a critical item are rated to have "low" overall confidence. Studies with more than one "no" in a critical item are rated to have "critically low" overall confidence

[30], so SRs that used this tool without modification received a "partial yes" at best on this criterion. On assessing confounding, NOS requires pre-specification of the two most important factors to control to satisfy the criteria of "comparability," which 6 out of the 8 SRs using the tool failed to specify [7, 20, 22, 23, 25, 27]. Selection bias as well as confounding were equivocally evaluated by most SRs due to ambiguity around the definition of control groups (see Table 1 and S5 Table). Most of the seven prevalence SRs that included the general adult population did not define a control group; one defined the control group as COVID-positive without post-discharge symptoms [20]; one defined it as people without COVID-19 but did not report the number of studies that used a control group or any associated outcomes [22]. This exposes interpretative challenges in how SRs applied the NOS criteria on "selection of non-exposed cohort" and "comparability." [30].

Risk factor SRs had lower ROB overall, with two SRs with 0–1 deficiency. All had clearly defined non-exposed comparator groups by the PICO criteria (lacking in the risk factor rather than COVID-19 exposure). Nevertheless, a majority were deficient on items 7 and 10, and 40% were deficient on items 9 and 11 (see Table 2).

## Discussion

This umbrella review found a wide range in the prevalence estimates of Long COVID primary studies, yielding pooled prevalence estimates that cluster around 50% (Fig 2), which needs to be interpreted in light of a major limitation. The presence of high heterogeneity demands the use of random-effects meta-analysis as was done in all the SRs reviewed [37]. But when between-study variance greatly exceeds within-study variance, as is the case when the $I^2$ statistic exceeds 90%, each primary study is given similar weight and the pooled estimate approximates the arithmetic mean [38]. It is thus no surprise that pooled prevalence estimates cluster around 50% when prevalence estimates in the primary literature spans nearly the entire range of numerical possibility (Fig 3). This may also explain why, with few exceptions, meta-analytic estimates of Long COVID prevalence consistently exceeds estimates from population-based samples [3, 39–41]. For instance, the June 7–19, 2023 wave of the U.S. Household Pulse Survey, which periodically samples a representative group of U.S. adults, suggested that Long COVID prevalence was 11.0% (95% CI: 10.4–11.6%) among U.S. adults reporting previous COVID-19, lower than any of the pooled prevalence estimates we reviewed [42].

Nevertheless, random-effects meta-analysis may be fruitfully applied to subgroups prespecified by study design and population characteristics. For instance, one SR observed a difference in prevalence estimates by hospitalization status, with lower prevalence estimates in studies of exclusively non-hospitalized compared to post-hospitalization cohorts [6]. The inclusion of control group and population sampling also generated lower prevalence estimates, although no meta-analysis was conducted in the only SR we included which reported outcomes by the use of these methods [21]. A recent SR not included in the date-range of our search estimated Long COVID absolute risk difference in community-based samples using control groups to be 10.1% (95% PI: -12.7%-32.8%) compared a pooled prevalence of 42.1% (95% PI: 6.8–87.9%) for all studies with more than 12 weeks of follow-up [41]. Timing of follow-up did not appear to significantly modify prevalence estimates in the four studies that reported on prevalence before and after 3 months of follow-up [6, 7, 20, 24], although overlaps in follow-up durations and inconsistent reporting in the primary literature may confound these findings. No SR specified enough subgroups to adequately address the range of factors likely contributing to high heterogeneity.

The risk factor SRs did not suffer as much from high heterogeneity. More than one SR discerned significant associations between increased COVID-19 risk and less than 2 pre-infection

vaccinations, female sex, and multiple comorbidities. The association between Long COVID risk and acute-COVID-19 hospitalization and severity were also significant in at least one SR. However, hospitalization and acute COVID-19 severity are strongly associated with selection into Long COVID studies, so one should be wary of spurious associations that emerge from collider bias, as has been demonstrated in other risk associations derived from test-positive or hospitalization-based COVID-19 cohorts [43].

Considering that SRs, coupled with meta-analyses, form the "capstone" of evidence-based medicine and public health [44], it is troubling that this review exposed a high level of ROB among prevalence SRs. Selection and measurement biases were reported across SRs. In prevalence SRs, bias towards hospitalized patients and survey respondents likely led to an over-estimation of Long COVID prevalence. Measurement bias, particularly the use of self-reporting without a standardized scale or blind independent assessment, was another recurrent ROB. Different approaches to Long COVID outcome assessment have been shown to produce prevalence estimates that vary from 3.0% based on tracking specific symptoms to 11.7% based on self-classification within the same sample population [45]. Our recommendations to address these and other sources of bias are elaborated in Table 3.

**Table 3. Recommendations for mitigating biases in Long COVID studies.**

|  | Primary Studies | Reviews |
|---|---|---|
| **Confounding** | Document common confounders including age, sex, comorbidities, race/ethnicity, severity of acute infection, hospitalization status, timing of infection, duration of follow-up, variants of concern, and pre-infection vaccination status | Define confounders in study protocol for data extraction, ROB assessment and subgroup/sensitivity analysis |
|  | Exploit EHR and ongoing cohort studies with robust data on patient characteristics that cannot be prospectively measured at the time of infection (e.g., pre-infection vaccination status, predominant variant of concern, pre-infection patient phenotype) | Use or revise an ROB tool to enable precise definition of the most important confounders a study should be designed to address |
|  | Include comparator/control cohort of people without COVID-19 enrolled concurrently with SARS-CoV-2 positive individuals | Conduct subgroup or sensitivity analysis as specifically as the number of studies and participants allow (e.g., not just hospitalized vs. non-hospitalized but also hospitalized with comparator cohort vs. non-hospitalized with comparator cohort) |
| **Selection bias** | PICO criteria of population and comparator group should be clearly specified and outcome-dependent. For example, methods of measuring long COVID based on periodic symptom-monitoring or electronic health records should have exposure-negative control groups; patient self-report of long COVID as the outcome should prioritize representative sampling to ensure inclusion of people with different healthcare access and education. | PICO criteria of the review should be clearly specified. PICO definition of the primary studies should be extracted for each study. Specify the applicability and comparability of the non-exposed cohort for every study in data extraction. |
|  | Recruit patients consecutively and report response rate or attrition for both case and comparator cohorts* | Differences in PICO criteria used in the primary studies should inform subgroup/sensitivity analysis. For instance, studies that did and those that did not use a comparator cohort should be analyzed as separate subgroups |
|  | Use PSM or IPW for matching cases with comparators especially if regression analysis is planned | Use an ROB tool with clear metrics for assessing sample selection or the suitability of comparator groups such as NOS or ROBINS-E |
| **Measurement bias** | Clearly define and document time zero and duration of follow-up, including range, measures of central tendency and variation* | Extract timing and duration of follow-up for all studies and conduct subgroup analysis accordingly as the number of studies and participants allow |
|  | Define or corroborate exposure status with confirmatory testing using RT-PCR or test of similar sensitivity and specify* | Extract method of exposure assessment for all studies and conduct subgroup analysis accordingly as the number of studies and participants allow |
|  | Administer established symptom scales such as PROMIS or EuroQol, ideally with both case and comparator groups, rather than relying on self-report of long COVID while the condition does not have a clear symptom-based definition | Extract method of outcome assessment (i.e., interview, survey, electronic health records) and conduct subgroup analysis accordingly as the number of studies and participants allow |

(*Continued*)

**Table 3.** (Continued)

| | Primary Studies | Reviews |
|---|---|---|
| **Outcomes selection** | Register protocol to include the tool and method used to ascertain Long COVID or a set of signs and symptoms that may qualify as Long COVID* | Specify the case definition used to include the set of symptoms and conditions measured |
| | Refer to consensus definitions supplemented by existing definitions for related illnesses including ME/CFS or other post-viral conditions* | Report study-level differences in the types of outcomes selected and aggregated to represent Long COVID |
| | Report symptom severity and degree of functional impairment experienced by patients* | Specify which outcomes of which studies are included in composite outcomes such as "at-least 1 symptom," including underlying sample size and population characteristics |

EHR, electronic health record; IPW, inverse probability weighing; ME/CFS, myalgic encephalomyelitis/chronic fatigue syndrome; NOS, Newcastle Ottawa Scale; PICO, population, intervention, comparator, outcome; PROMIS, Patient-Reported Outcomes Measurement Information System; PSM, propensity score matching; ROB, risk of bias; ROBINS-E, Risk of Bias In Non-randomized Studies—of Exposure; RT-PCR, reverse transcriptase polymerase chain reaction

*: Adapted from *Nasserie T*, *Hittle M*, *Goodman SN. Assessment of the Frequency and Variety of Persistent Symptoms Among Patients With COVID-19*: *A Systematic Review. JAMA Netw Open. 2021;4(5):e2111417*

A major limitation of this review is that the reporting of study and population characteristics of the primary literature in the SRs reviewed, including the ROB assessments, lacked sufficient consistency, granularity, and methodological transparency. Aggregating across a kitchen-sink metric like "at least one symptom" when what counts as a Long COVID symptom differs across the primary studies obviously hinders measurement and interpretation. Yet, we had little choice but to use this outcome as it is a widely accepted operational definition of Long COVID. An overall high ROB of the SRs corresponds to low certainty in our outcome estimates. Nevertheless, consistent themes emerged in the ROB assessments and limitations reviewed.

Our review highlights four major areas of limitation and bias in Long COVID observational studies: 1) few primary studies used techniques of representative sampling or included non-exposed comparator cohorts; 2) both primary studies and SRs lacked uniformity and consistency in reporting potential confounders, including factors that may now be impossible to prospectively measure (e.g., pre-vaccination SARS-CoV-2 exposure); 3) a high overall ROB in the SRs, including inadequate ROB assessment of the primary studies; 4) primary studies and SRs selected a wide variety of outcomes to measure, contributing to high heterogeneity when aggregating across studies. A clear and consistent research definition of Long COVID with corresponding protocols for measurement would be an important intervention to reduce heterogeneity across Long COVID studies. The National Academies of Science, Engineering and Medicine have been tasked to examine the current U.S. government working definition of Long COVID, the culmination of which could bring much-needed standardization in Long COVID research [46]. However, this would only mitigate the fourth limitation, while the first three depend on improving study quality independent of heterogeneity stemming from an inconsistent case definition. In Table 3, we augment an existing set of recommendations [24] for improving uniformity in the Long COVID primary literature and address sources of bias in the review literature. The effort to develop and maintain quality standards for measuring and monitoring Long COVID is not only important for understanding the long shadow of COVID-19, but in preparation for tracking post-infective conditions of future novel pathogens.

## Supporting information

**S1 Table. Database search strategies.**
(DOCX)

**S2 Table. Articles excluded after full-text review.**
(DOCX)

**S3 Table. Study publication and design information.**
(DOCX)

**S4 Table. Summary of statistical outcomes, hospitalization status and follow-up time points.**
(DOCX)

**S5 Table. Comparator groups and subgroups defined by vaccination status and variants of concern.**
(DOCX)

**S6 Table. PRISMA 2020 checklist.**
(DOCX)

## Author Contributions

**Conceptualization:** Miao Jenny Hua, Gisela Butera, Deborah Porterfield.

**Data curation:** Miao Jenny Hua, Gisela Butera.

**Formal analysis:** Miao Jenny Hua.

**Investigation:** Miao Jenny Hua, Oluwaseun Akinyemi, Deborah Porterfield.

**Methodology:** Miao Jenny Hua, Gisela Butera, Deborah Porterfield.

**Project administration:** Deborah Porterfield.

**Resources:** Deborah Porterfield.

**Software:** Miao Jenny Hua, Gisela Butera.

**Supervision:** Deborah Porterfield.

**Validation:** Oluwaseun Akinyemi.

**Visualization:** Miao Jenny Hua.

**Writing – original draft:** Miao Jenny Hua.

**Writing – review & editing:** Miao Jenny Hua, Deborah Porterfield.

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
