## [Decision Letter · Decision Letter 0]

16 Feb 2024

PONE-D-23-43411Biases and Limitations in Observational Studies of Long COVID Prevalence and Risk Factors: A Rapid Systematic Umbrella ReviewPLOS ONE

Dear Dr. Butera,

Thank you for submitting your manuscript to PLOS ONE. After careful consideration, we feel that it has merit but does not fully meet PLOS ONE’s publication criteria as it currently stands. Therefore, we invite you to submit a revised version of the manuscript that addresses the points raised during the review process.

This manuscript about biases and limitations in observational studies of Long COVID provides a wide vision about the problem of trying to make reviews based on observational studies with very heterogeneous methodologies and diagnostic criteria. This is an important reflection and I thank you for the submission to our journal.

Please, find attached the reviewers' comments, which may help to improve some aspects of your paper.

We look forward to receiving your revised manuscript.

Kind regards,

Paulo Alexandre Azevedo Pereira Santos, PhD

Academic Editor

PLOS ONE

2. PLOS requires an ORCID iD for the corresponding author in Editorial Manager on papers submitted after December 6th, 2016. Please ensure that you have an ORCID iD and that it is validated in Editorial Manager. To do this, go to ‘Update my Information’ (in the upper left-hand corner of the main menu), and click on the Fetch/Validate link next to the ORCID field. This will take you to the ORCID site and allow you to create a new iD or authenticate a pre-existing iD in Editorial Manager. Please see the following video for instructions on linking an ORCID iD to your Editorial Manager account: https://www.youtube.com/watch?v=_xcclfuvtxQ.

Additional Editor Comments:

This manuscript about biases and limitations in observational studies of Long COVID provides a wide vision about the problem of trying to make reviews based on observational studies with very heterogeneous methodologies and diagnostic criteria. This is an important reflection and I thank you for the submission to our journal.

Please, find attached the reviewers' comments, which may help to improve some aspects of your paper.

Reviewers' comments:

Reviewer's Responses to Questions

**Comments to the Author**

1. Is the manuscript technically sound, and do the data support the conclusions?

Reviewer #1: Yes

Reviewer #2: Yes

2. Has the statistical analysis been performed appropriately and rigorously? 

Reviewer #1: Yes

Reviewer #2: Yes

3. Have the authors made all data underlying the findings in their manuscript fully available?

Reviewer #1: Yes

Reviewer #2: Yes

4. Is the manuscript presented in an intelligible fashion and written in standard English?

Reviewer #1: Yes

Reviewer #2: Yes

5. Review Comments to the Author

Reviewer #1: This umbrella review provides and interesting focus on biases and limitations. Below are some methodological questions I would like the authors to address.

* What was “rapid” about this review, and which corners were cut (eg, “components of the methodology may be simplified or omitted” on line 73–74) by not doing a “standard” umbrella review? Not only omitting searches of grey literature I assume. Which are the limitations of this approach?

* The authors assessed quality in terms of AMSTAR scores (risk of bias) but other quality tests are missing, including small-study effects and excess significance bias (see for example: https://pubmed.ncbi.nlm.nih.gov/37898519/). I suggest the authors include these additional measures of quality – or at least mention in the discussion that these were not examined as quality assessment was based only on AMSTAR scores (which is only a partial measure).

* Could the authors also include a (short) discussion (and figure) of high-quality (low risk of bias) studies only? This would be informative to see what the ‘best’ evidence demonstrates in terms of prevalence and risk factors.

Reviewer #2: This umbrella review synthesized estimates of Long COVID prevalence and risk factors, while also analyzing methodological challenges in combining data from observational studies. Fourteen reviews, covering 5-196 primary studies, were included, revealing a prevalence range of Long COVID symptoms from 21% to 74.5%. Significant associations were found between Long COVID risk and factors such as vaccination status, sex, acute COVID-19 severity, and comorbidities. However, both prevalence and risk factor reviews identified biases, and the quality of reviews, particularly in prevalence estimates, raised concerns regarding bias assessments and meta-analysis justifications. The interpretation of pooled prevalence estimates is challenging due to heterogeneity and lack of robust critical appraisals. Nonetheless, risk factor reviews indicated consistent associations between Long COVID risk and patient characteristics. This article is highly important and relevant. However, further revisions are recommended to refine the manuscript:

1. In the introduction, it is crucial to emphasize that long COVID symptoms following an acute SARS-CoV-2 infection can persist for nearly two-thirds of individuals who have had an acute SARS-CoV-2 infection, as referenced in the following papers: DOI: 10.1002/jmv.28852, DOI: 10.1016/j.jinf.2023.12.004.

2. Kindly include a table showcasing the different keywords and combinations used for the search strategy.

3. For Tables 1 and 2, it appears that there were articles cited that were not included in the references section. Kindly double-check to ensure that all citations are referenced appropriately.

6. PLOS authors have the option to publish the peer review history of their article (what does this mean?). If published, this will include your full peer review and any attached files.

Reviewer #1: No

Reviewer #2: No

---

## [Author Response · Author response to Decision Letter 0]

14 Mar 2024

Reviewer #1 

* What was “rapid” about this review, and which corners were cut (eg, “components of the methodology may be simplified or omitted” on line 73–74) by not doing a “standard” umbrella review? Not only omitting searches of grey literature I assume. Which are the limitations of this approach?

We elaborated on how small changes to the standard umbrella review protocol ensured rapidity without compromising on quality on p. 5, lines 76-79: “This review omitted searches of grey literature and data extraction was performed by a single reviewer, which expedited the review process to under six months without compromising on other areas of a systematic review (e.g., critical appraisal) felt to be crucial to ensuring an unbiased protocol.”

* The authors assessed quality in terms of AMSTAR scores (risk of bias) but other quality tests are missing, including small-study effects and excess significance bias (see for example: https://pubmed.ncbi.nlm.nih.gov/37898519/). I suggest the authors include these additional measures of quality – or at least mention in the discussion that these were not examined as quality assessment was based only on AMSTAR scores (which is only a partial measure).

The additional measures of quality Reviewer #1 recommends, such as statistical measures of small-study effect, are intended for discerning bias in the outcomes of meta-analyses. We have decided not to conduct a meta-analysis due to the high level of heterogeneity in the underlying studies, as we explored extensively in the Discussion section of this article, so these additional quality tests are not applicable. AMSTAR 2 is a critical appraisal tool based on the well-validated AMSTAR adapted for the assessment of both randomized and non-randomized studies. It is thus a comprehensive tool for one of the main objectives of this study, which was to discern limitations and biases in the literature of Long COVID observational studies.

* Could the authors also include a (short) discussion (and figure) of high-quality (low risk of bias) studies only? This would be informative to see what the ‘best’ evidence demonstrates in terms of prevalence and risk factors. 

There were only 2 studies which were of high quality (Byambasuren et al. & Pillay et al.), both of which are what we called “Risk Factor SRs” but they explored different outcomes through different methods (e.g., Pillay et al. conducted meta-analyses on select outcomes while Byambasuren et al. did not at all). Because we also did not conduct a meta-analysis, the results could not be further synthesized into a figure without distortions. The outcomes of the individual studies have been extracted and summarized in Supplemental Table 4. Discussion on the relative higher quality of Risk Factor SRs are included on p. 14 lines 274-7 and p. 16 lines 309-16.

Reviewer #2 

1. In the introduction, it is crucial to emphasize that long COVID symptoms following an acute SARS-CoV-2 infection can persist for nearly two-thirds of individuals who have had an acute SARS-CoV-2 infection, as referenced in the following papers: DOI: 10.1002/jmv.28852, DOI: 10.1016/j.jinf.2023.12.004.

We appreciate the reviewer’s suggestion of further references. The first article linked (DOI: 10.1002/jmv.28852) is included in our systematic review (Rahmati et al. 2023). It does not indicate that Long COVID symptoms persist for >50% of individuals after acute COVID-19 (the outcome we extracted is 41.7%). The second article (DOI: 10.1016/j.jinf.2023.12.004; Fernandez-de-Las-Peñas et al., 2024) states that Long COVID symptoms persist for around 30% of individuals after acute COVID-19. 

However, in the spirit of Reviewer #2’s comment, we have included a reference to an early, well-cited, systematic review that suggested as many as 80% of COVID-19 survivors experience long-term symptoms (p. 4, lines 50-2).

2. Kindly include a table showcasing the different keywords and combinations used for the search strategy.

Full database search strategies including all keyword combinations used were included in Supplemental Table 1. 

3. For Tables 1 and 2, it appears that there were articles cited that were not included in the references section. Kindly double-check to ensure that all citations are referenced appropriately.

One reference (Notarte et al.) was accidentally omitted. We have corrected the references in this version of the manuscript and summarized below:

Study Reference Number

Byambasuren et al. 26

Di Gennaro et al. 7

Fernandez et al. 19

Huang et al. 20

Ma et al. 23

Nasserie et al. 24

Nittas et al. 21

Notarte et al. 29

O'Mahoney et al. 6

Pillay et al. 28

Rahmati et al. 18

Tsampasian et al. 25

Watanabe et al. 27

Zeng et al. 22

---

## [Editor Report · Decision Letter 1]

3 Apr 2024

Biases and Limitations in Observational Studies of Long COVID Prevalence and Risk Factors: A Rapid Systematic Umbrella Review

PONE-D-23-43411R1

Dear Dr. Butera,

We’re pleased to inform you that your manuscript has been judged scientifically suitable for publication and will be formally accepted for publication once it meets all outstanding technical requirements.

Kind regards,

Paulo Alexandre Azevedo Pereira Santos, PhD

Academic Editor

PLOS ONE